

# The importance of local settings: within-year variability in seawater temperature at South Bay, Western Antarctic Peninsula

César A. Cárdenas, Marcelo González-Aravena and Pamela A. Santibañez

Departamento Científico, Instituto Antártico Chileno, Punta Arenas, Chile

## ABSTRACT

The Western Antarctic Peninsula (WAP) has undergone significant changes in air and seawater temperatures during the last 50 years. Although highly stenotherm Antarctic organisms are expected to be severely affected by the increase of seawater temperature, high-resolution datasets of seawater temperature within coastal areas of the WAP (where diverse marine communities have been reported) are not commonly available. Here we report on within-year (2016–2017) variation in seawater temperature at three sites on Doumer Island, Palmer Archipelago, WAP. Within a year, Antarctic organisms in South Bay were exposed to water temperatures in excess of 2 °C for more than 25 days and 2.5 °C for more than 10 days. We recorded a temperature range between −1.7° to 3.0 °C. Warming of seawater temperature was 3.75 times faster after October 2016 than it was before October. Results from this study indicate that organisms at South Bay are already exposed to temperatures that are being used in experimental studies to evaluate physiological responses to thermal stress in WAP organisms. Continuous measurements of short to long-term variability in seawater temperature provides important information for parametrizing meaningful experimental treatments that aim to assess the local effects of environmental variation on Antarctic organisms under future climate scenarios.

## INTRODUCTION

Although there is an absence of continent-scale warming over Antarctica over the last 100 years (*Stenni et al., 2017*), during the last 50 years the Western Antarctic Peninsula (WAP) has undergone significant warming in air and seawater temperatures, with the later increasing by ∼2 °C (*Cook et al., 2016*; *Meredith & King, 2005*; *Stenni et al., 2017*). Warming has contributed to the range expansion of various marine flora (e.g., *Convey & Smith, 2006*; *Smith, 1994*; *Hill et al., 2011*), the loss of glacier mass, the retreat of ice shelves, and summer melting, with the latter being unprecedented over the past 1,000 years (*Abram et al., 2013*; *Paolo, Fricker & Padman, 2015*). Increased snowfall on the Antarctic Peninsula has been linked to warming sea surface temperatures in the western Pacific (*Thomas et al., 2015*; *Thomas et al., 2017*) and the warming of WAP mid-ocean depths (rather than

Corresponding author
César A. Cárdenas,
ccardenas@inach.cl

the warming of air temperatures) has been proposed as the primary driver of glacier retreat (e.g., *Cook et al., 2016*). These changes highlight the complexity of environmental change occurring at the WAP and the need for high-resolution information on regional-to local-scale patterns of climate change, to address its implications for marine polar ecosystems (e.g., *Clarke et al., 2007*; *Meredith, Stefels & Van Leeuwe, 2017*).

Many Antarctic species are not well adapted to sudden temperature change because they have evolved in one of the most stable environments on Earth (e.g., *Peck, 2005*; *Morley et al., 2016*; *Peck, Webb & Bailey, 2004*). Some Antarctic species are sensitive to even small changes in temperature (2–3 °C above the annual average), losing the ability to perform essential functions (*Ingels et al., 2012*; *Peck, Morley & Clark, 2010*; *Peck, Webb & Bailey, 2004*; *Peck, 2005*; *Pörtner, Peck & Somero, 2007*). For example, the Antarctic scallop *Adamussium colbecki* loses its ability to perform essential activities (*Peck, 2005*) with increases in seawater temperature of 2−3 °C, and attempts to acclimate individuals to temperatures above 2 °C have failed for this species (*Bailey, Johnston & Peck, 2005*). Experiments using the Antarctic clam *Laternula elliptica*, have shown that half of the individual died in 60 days at 3 °C, and for the brittle star *Ophionotus victoriae* half of the individuals died in less than 30 days at 3 °C (*Turner et al., 2009*). In contrast others organisms, such as the sea urchin *Sterechinus neumayeri,* are comparatively resistant to increases in seawater temperature (*Ericson et al., 2012*; *Suckling et al., 2015*). Recent studies have highlighted the dramatic effects of warming waters on Antarctic organisms at the community level (*Ashton et al., 2017*), where small increases in temperature (1 °C) altered community composition by reducing overall species diversity and evenness. The study of resistance, adaptation and tolerance to environmental variation has become a key focus of evaluating how Antarctic ecosystems respond to climate change (*Gutt et al., 2012*; *Ingels et al., 2012*; *Peck, 2005*). Research to date has focused on identifying how Antarctic organisms respond to the expected seawater temperature increase at community (*Ashton et al., 2017*), assemblage (*Peck et al., 2009*) and species (physiological and molecular) levels (*Cascella et al., 2015*; *Ericson et al., 2012*; *Foo et al., 2016*; *González et al., 2016*; *Kapsenberg & Hofmann, 2014*; *Peck et al., 2009*; *Peck et al., 2014*; *Schram et al., 2015*; *Suckling et al., 2015*), with the latter being the most studied. To inform the design of studies that are relevant to future climate scenarios high-resolution records of local to regional patterns of water temperature are required.

Due to the physical constraints that the Antarctic continent imposes on equipment and sampling, obtaining long-term high-resolution water temperature data is often logistically challenging. The urgent need to increase the spatio-temporal coverage of surface seawater temperature data, with a focus on coastal areas, has been highlighted by several authors (*Gutt et al., 2017*; *Stenni et al., 2017*). Although a number of moorings have been deployed along channels and open waters in the WAP (See http://www.soos.aq/activities/soos-at-sea/moorings), there remains a lack of long-term, high-resolution temperature data from shallow waters in coastal zones dominated by diverse communities of sessile organisms with limited behavioral thermoregulatory ability (*Ashton et al., 2017*). Fine-scale temporal (daily) data can be crucial for inferring climatic impacts on survival, growth and reproduction (*Kearney, Matzelle & Helmuth, 2012*), while monthly data is better suited

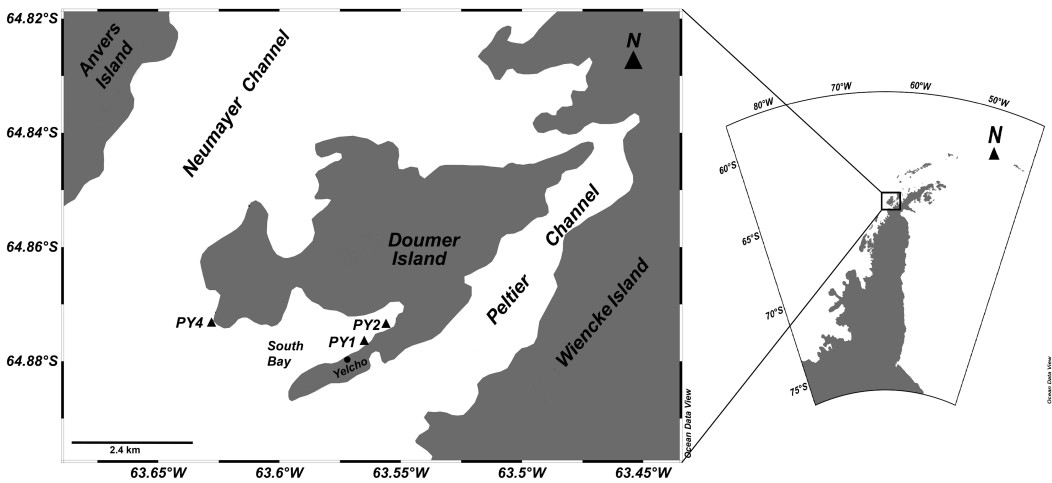

**Figure 1** **Study site.** Locations of HOBO temperature/light data loggers deployments for studying sponge assemblages at Doumer Island (Palmer, Archipelago), WAP.

to studying relationships between population dynamics and climatic conditions (*Gutt et al., 2017*; *Jan et al., 2017*; *Kennicutt et al., 2016*). Observational gaps in surface seawater temperature leave fractionated information that does not provide a coherent and robust picture of the current and future conditions occurring in Antarctic ecosystems. In order to fill some of these gaps, high-resolution temporal observations are required to advance ecosystem modelling and projections (*Gutt et al., 2017*).

Here we report for the first time extreme changes in shallow seawater temperature around Doumer Island, Palmer Archipelago (WAP). We present shallow seawater temperatures recorded over one year at high temporal resolution (hours), which can be used to parameterize future physiological experiments.

## MATERIAL AND METHODS

This study was conducted from the Chilean Yelcho Scientific Station in South Bay, Doumer Island, Palmer Archipelago, WAP (Fig. 1), where diverse benthic communities have been described (*Cárdenas et al., 2016*; *Zamorano, 1983*) (Fig. 2). Three HOBO pendant temperature/light data loggers (Part #UA-002-XX; Onset Computer Corp; accuracy: ±0.5 °C at 0 °C, resolution: 0.1 °C at 0 °C) were deployed in January 2016 and recovered in February 2017 at three sites in South Bay (Fig. 1). Sites were separated by ∼1.5 km and are part of a research project assessing the responses of Antarctic sponge assemblages to climate change. Data loggers from two of the three sites (PY1; 64°52′28.1″S; 63°34′36.0″W and PY4; 64°51′58.6″S; 63°37′46.7″W) were at 10 m depth, whereas the third data logger (PY2; 64°52′15.3″S; 63°33′52.2″W) was at a depth of 20 m. Data was recorded every 2 h at sites PY2 and PY4, and every 1 h at PY1. Sites PY1 and PY2 were located inside the bay and were characterized by dense beds of *Himantothallus grandifolius* with a sub-canopy of crustose calcareous algae and sponges on a boulder/gravel substrate (Fig. 2). PY4 was located at Cape Kemp, which is regularly exposed to swell and is dominated by a mix of *H. grandifolius*,

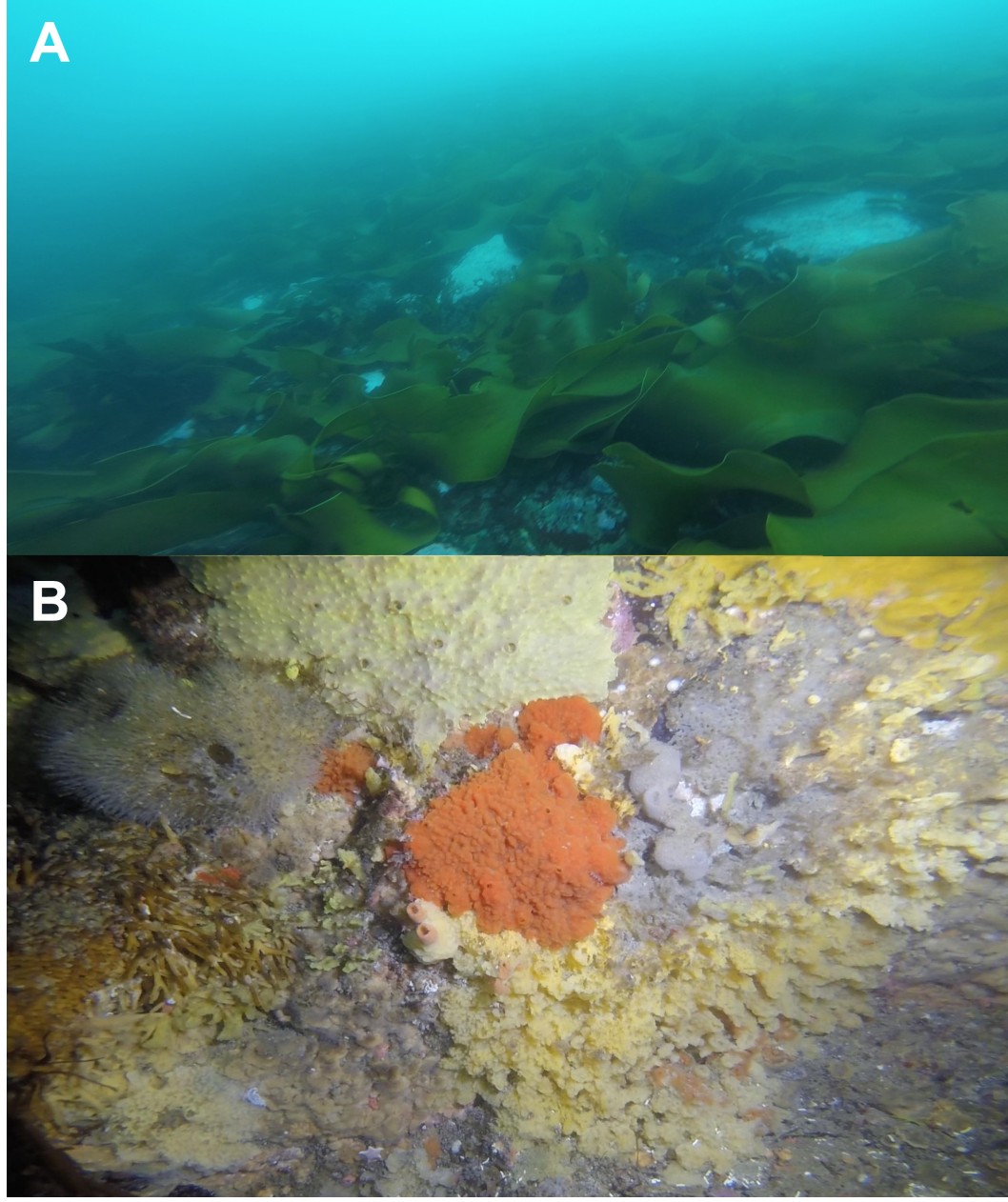

**Figure 2  Benthic communities at Doumer Island (Palmer Archipelago), WAP.** (A) algae-dominated zones (10 m depth), where (B) diverse sponge assemblages have been identified underneath the algal canopy. The area showed in (B) is approximately 80 cm × 50 cm.

*Desmarestia* spp. and *Ascoseira mirabilis* on a bedrock substrate. Benthic coverage of sponges at PY4 was higher than in sites inside the bay (*Bravo, 2017*). The data loggers at PY1, PY2 and PY4 recorded data for 404, 374 and 312 days, respectively. Data from PY4 stopped by late November (2016), presumably after being damaged by an iceberg. The study was conducted under the permit 806/2015 granted by the Chilean Antarctic Institute (INACH).

In order to describe non-linear, site-specific variability in the within-year patterns of seawater temperature, we fit separate Generalized Additive Models (GAMs) to each of the three sites, as they do not prescribe any particular form for the trend (*Ferguson et al., 2008*; *Hastie & Tibshirani, 1990*; *Orr et al., 2015*). We chose GAMs instead of other techniques (e.g., linear regression) because: (1) GAMs do not assume linearity and allow specification of the mean value of the response and the systematic component using a link-function (*Wood, 2006*; *Zuur, Ieno & Smith, 2007*); (2) GAMs allow the application of an autoregressive structure to account for dependency of serial correlations in time series data; and (3) GAMs allow the use of thin-plate shrinkage smoothers to model temporal trends (recommended in *Wood, 2006*). The estimated smooth trends are described by: (i) the shape of the estimated curve, (ii) a hypothesis test for no variation over time, and (iii) *edf*, which describes effect complexity and corresponds to the amount of information used by each smoother. An *edf* of 0 indicates no variation, an *edf* of 1 indicates a linear relationship between the response and the covariate, and higher values correspond to more complex non-linear relationships; and (4) GAMs allow the use of finite differences (*Curtis & Simpson, 2014*; *Monteith et al., 2014*) to estimate periods where the rate of change (slope) is statistically different from zero (i.e., periods where the temperature is either increasing or decreasing in a significant manner). This was achieved by computing the first derivatives of the fitted trend. Uncertainty estimates were also calculated for the derivatives to form approximate 95% confidence intervals.

All models were fit using the R-programming language (version 3.3.3 by R Foundation for Statistical Computing, 2017) and the package mgcv (*Wood, 2004*; *Wood, 2006*). The R code and scripts are available in the Supplemental Information.

## RESULTS

All seawater temperature records showed significant seasonality ($p$-value < 0.01; Fig. 3), with clear minima in winter (July–September) and maxima in summer (January–March). Although data loggers were located at different sites and depths (Fig. 1) they showed a similar seasonal pattern throughout the year (Fig. 4), implying that the three sites are representative of broader shallow waters temperatures in South Bay. In addition, sea surface temperatures within the same time frame from Port Arthur (Palmer Station, ~25 km from Yelcho Station) show similar temperature ranges and patterns suggesting that the temperature variability reported could be representative of the wider Palmer Archipelago (Fig. 4). Mean daily temperatures ranged from −1.7° to 3.0 °C (range = 4.7 °C, $SD = 1.1$) at PY1 (the longest record, 404 days), from −1.6° to 1.0 °C (range = 2.6 °C, $SD = 0.8$) at PY4 (the shortest record, 312 days), and from −1.8° to 1.8 °C (range = 3.6 °C, $SD = 0.8$) at PY2. Seawater temperature differed between the 2016 (max 0.8 °C) and 2017 (max 3.0 °C) summers at PY1 ($p$-value < 0.01; Fig. 5 and Fig. S1).

To identify periods of statistically significant change in the time series of seawater temperature at PY1 we calculated the first derivative method from a second GAM model ($k = 58$, $edf = 55.7$, $p$-value < 0.001, Fig. 5 and Fig. S2), which allowed a closer examination of the temperature variability at PY1 site. Figure 5 shows there were 26 periods of statistically

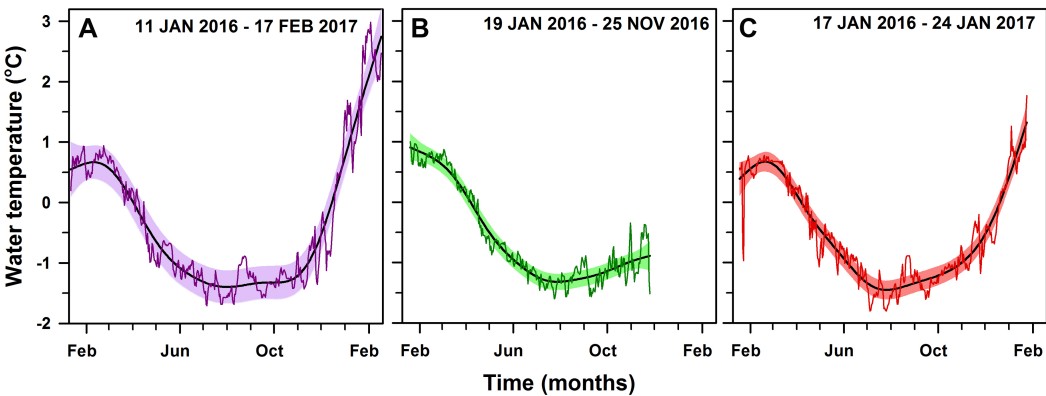

**Figure 3** **Daily mean seawater temperature.** Within-year GAM (solid line) of daily mean seawater temperature (°C) in (A) PY1-10 m ($edf = 6.7$, $p$-value < 0.001, 404 days), (B) PY4-10 m ($edf = 5.7$, $P < 0.001$, 312 days) and (C) PY2-20 m ($edf = 7.1$, $P < 0.001$, 374 days), respectively, at South Bay, Doumer Island, WAP. Black solid curves are the fitted seasonal model from the GAM. Colored areas correspond to 95% confidence bands and the colored thin lines are the observational data (temperature daily means).

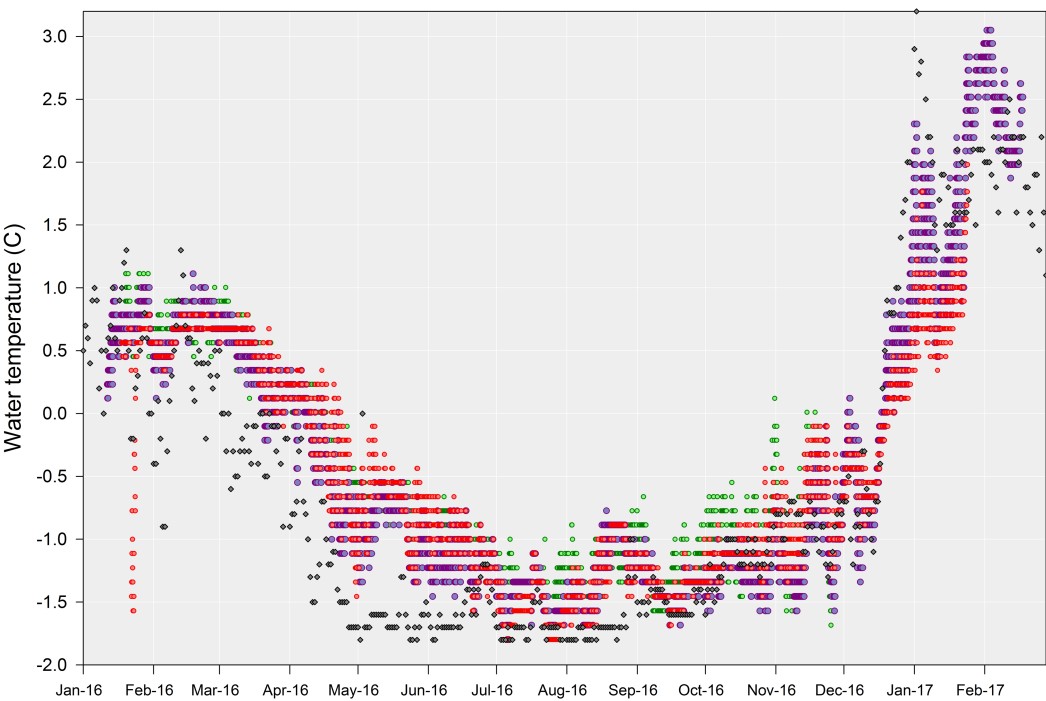

**Figure 4** **Shallow sea water temperatures recorded at Doumer Island.** Full records of shallow sea water temperatures recorded every 2 h at PY2-20 m (red dots) and PY4-10 m (green dots), and every 1 h at PY1-10 m (violet dots). Grey diamonds represent sea surface temperature average for the day, taken from Port Arthur (Palmer Station) in Anvers Island (http://pal.lternet.edu).

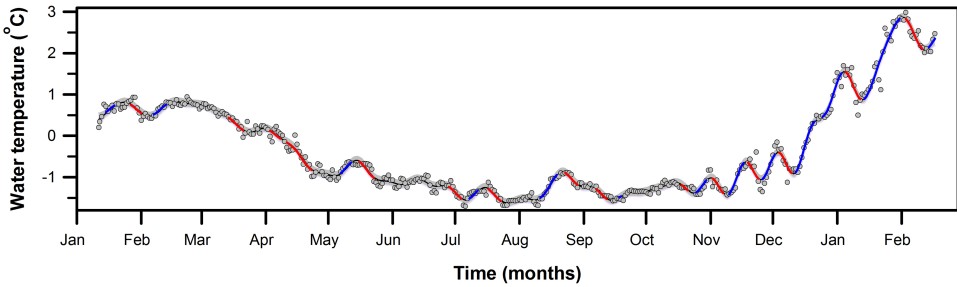

**Figure 5** **First derivative of seawater temperature trend from PY1 (10 m).** Fitted GAM trend ($k = 58$) showing estimate time intervals with significant rates of change in seawater temperature. Periods of significant change in the fitted trend, determined by derivative analysis, are highlighted by the thickened colored sections of the trend (red = decrease; blue = increase). Grey areas around the black line (fitted trend) indicate 95% confidence limits, and the dots are the daily mean seawater temperature recorded at PY1 from January 11th, 2016 to February 17th, 2017 in South Bay, Doumer Island, WAP.

**Table 1** **Periods of significant temperature increase.** Calculated warming rates (°C day-1) of periods of significant temperature increase from PY1 fitted GAM ($k = 58$, $edf = 55.7$, $p$-value $< 0.001$, Fig. 2).

| Periods of significant temperature increase | Rates (°C day$^{-1}$) |
| --- | --- |
| 15–19 Jan 2016 | 0.04 |
| 7–13 Feb 2016 | 0.04 |
| 7–13 May 2016 | 0.05 |
| 8–12 Jul 2016 | 0.05 |
| 11–19 Aug 2016 | 0.07 |
| 18–19 Sep 2016 | 0.03 |
| 25–30 Oct 2016 | 0.06 |
| 10–17 Nov 2016 | 0.12 |
| 26 Nov–2 Dec 2016 | 0.11 |
| 12–21 Dec 2016 | 0.15 |
| 25 Dec 2016–3 Jan 2017 | 0.13 |
| 14–31 Jan 2017 | 0.12 |

significant change throughout the year, 12 increases and 13 decreases. From all periods of significant increasing temperatures, the rate of temperature increase (°C day$^{-1}$) to which marine organisms were subjected varied between 0.11 to 0.15 °C day$^{-1}$ (mean 0.13, $SD = 0.01$) after October 2016. In contrast, the rate of increase before October 2016 ranged between 0.04 and 0.07 °C day$^{-1}$ (mean = 0.05, $SD = 0.02$; Table 1).

## DISCUSSION

In recent years, several studies have focused on assessing how Antarctic organisms will adapt to future environmental conditions (e.g., *Cascella et al., 2015*; *Ericson et al., 2012*; *Peck, 2011*; *Peck, 2016*; *Peck et al., 2014*; *Schram et al., 2015*). Existing studies have used different species, methodologies and warming rates, which sometimes make comparing species-specific responses and treatment effects difficult (*Clark et al., 2017*). In addition,
most experimental studies have used standard scenarios of climate change that are not necessarily relevant to nearshore areas that may be affected by a higher degree of temperature variability than other marine environments (*Vargas et al., 2017*). It is therefore important to incorporate realistic scenarios of temperature change into evaluations of ecosystem responses to climate change across multiple spatial and temporal scales (*Gutt et al., 2017*).

The present work provides the first yearlong record of shallow seawater temperatures around Doumer Island and one of the few on the Palmer Archipelago, WAP. Results show short-term variability and significant warming events. This constitutes important baseline information that could be used to assess long-term trends in local seawater temperature, and inform future physiological and molecular experimental studies in the area. Although HOBO data loggers have an accuracy of $\pm 0.5\,°C$ at $0\,°C$ which could increase the temperature range reported in our study, our results are in accordance with those obtained from 24-hour CTD measurements carried out in parallel by other researchers during certain days of the field season. Results demonstrate that marine organisms, at least in some coastal areas of Antarctica, are already exposed seawater temperatures that have been widely used in physiological tolerance experiments to evaluate the effects of future climate scenarios. For example, observed temperatures in this study of more than $2.0\,°C$ for more than 25 days or $2.5\,°C$ for more than ten days within a year, have been used experimentally to test responses to acute heat stress (e.g., *Peck et al., 2009*). Further, *Waller et al. (2017)* reported for the first time the presence of the bivalve *Laternula elliptica* (has been used as a model to test the responses of Antarctic organisms to warming sea temperatures) in intertidal habitats were sediment and air temperature reached more than 7 to $10\,°C$, respectively. That study (*Waller et al., 2017*) and the continuous records of water temperature reported here suggest that we may need to rethink the temperature ranges used in experimental setups to better reflect local conditions, as organisms occurring on different regions of Antarctica might respond differently to environmental variation.

Short-term variability in water temperature is relatively low in Antarctic coastal environments, with a mean annual temperature range of $\sim 3.1\,°C$ at some places such as Anvers Island (*Barnes et al., 2006*); hence, it is expected to have very little impact on ecology (*Clarke, 2001*). However, recent work highlights the importance of environmental variability at different scales, with acute thermal stress events and even timing of seasonal events being suggested as critical factors driving physiological, ecological and evolutionary processes (See *Fordham et al., 2017* and references therein). Although our time-series of continuous, directly-measured sea temperatures is limited to a year, it shows the complete annual pattern and existence of dynamic variations within South Bay, Doumer Island. Our results are in accordance with sea surface temperature data from Port Arthur, Palmer Station in Anvers Island (http://pal.lternet.edu/), which is approximately 25 km west of Doumer Island, suggesting this pattern occurs at least throughout the Palmer Archipelago (Fig. 4). Additionally, there are some long-term data sets at WAP for a few areas (e.g., Palmer LTER, Rothera Station-BAS Database) that allowed comparison.

Our results show that marine organisms at South Bay were exposed to a broad annual temperature range from $-1.7°$ to $3.0\,°C$ with a range of $4.7\,°C$. Antarctic marine organisms are recognized as the most stenothermal on Earth (i.e., organisms only capable of surviving

over a narrow range of temperature). For example, some species such as the brittlestar *Ophionotus victoriae* cannot tolerate acclimation temperatures of more than 2–3 °C (*Peck et al., 2009*). Based on an experimental approach using 3.5 °C seawater temperatures (representative of current summer transient peaks of sea surface recorded at Port Arthur, Anvers Island), *Schram et al. (2015)* suggested that an increase in the frequency and peaks of warning events may affect feeding preferences of amphipod species. Considering the observed within-year trends in water temperature recorded at South Bay, it is important to assess and monitor the potential effects on physiological and ecological processes due to rising water temperatures. The effects of increased seawater in sessile organisms, such as sponges, which form diverse assemblages in the shallows (*Cárdenas et al., 2016*), have not been well studied in Antarctica. Evidence from other latitudes suggests some sponge species might be highly vulnerable to small increases in seawater temperature (*Lopez-Legentil et al., 2011*), however this remains to be tested in Antarctic species.

Previous work has shown that even small variations in environmental condition can have significant effects on the physiology of some organisms (*Morley et al., 2010*). This is highly relevant considering thermal limits cannot only vary spatially but also between years and even seasons (*Morley et al., 2012*; *Morley et al., 2016*). Laboratory studies have used rapid warming rates, which are 100–10,000 times faster than those observed here and predicted for seawater temperatures (*Peck et al., 2009*). Our HOBO data loggers recorded a significant change in the warming rates before and after October 2016 (3.75 times higher after October 2016, Table 1, Fig. 4). Warming rates are positively correlated with Critical Thermal Maximum (CTMax), which is used to describe the effects of heat tolerance on different organisms (*Bilyk & DeVries, 2011*; *Cascella et al., 2015*; *Mora & Maya, 2006*; *Peck et al., 2009*; *Peck et al., 2014*; *Terblanche et al., 2007*). Lower CTMax results from slower warming rates and consequently poorer thermal tolerance. Our results suggest that caution must be applied when interpreting CTMax derived from laboratory experiments because some of those values were obtained by using warming rates that are far faster than that experienced by organisms in their natural habitats (in this study between 0.03 and 0.15 °C per day). In addition, differences in warming rates before and after October 2016 (0.05 [SD: 0.01] vs. 0.13 [SD: 0.02] °C day) can have a significant impact on the mechanisms dictating the survival limits of benthic organisms (see *Peck et al., 2009*).

Recent studies have shown the importance of including natural local variability (with several treatment levels) in experimental designs to avoid exposing organisms to unrealistic scenarios (*Vargas et al., 2017*). Local conditions should be taken into account since organisms occurring in areas where high variability occurs, might show different responses to stressors compared to those from areas where environmental variability is lower. For instance, it is possible that organisms inhabiting zones that have more stable temperatures (e.g., East Antarctica) might show different responses to warming than those inhabiting areas exposed to higher temperature variability such as the WAP. The use of continuous measurements of temperature and other environmental data constitute a valuable tool to identify long- and short-term variability in coastal waters. This type of information can provide more realistic scenarios based on local conditions in future studies assessing the effect of environmental variation on Antarctic organisms.

## CONCLUSIONS

The use of continuous measurements of environmental data, such as seawater temperature, can provide key information to identify current patterns of long- and short-term variability in Antarctic coastal waters. Such information is important in informing realistic scenarios for experimental work assessing the local effect of environmental variation on Antarctic organisms. Our results show that benthic organisms at South Bay were exposed to high local variability in seawater temperature (range: $-1.7°$ to $3.0°$), rates of warming (0.04 to 0.15 °C day$^{-1}$), and duration of elevated temperatures (more than 25 days over 2 °C). These results demonstrate that seawater temperatures at depths of 10 to 20 m during summer in South Bay already reach those predicted temperatures for 2100. Although our data was collected in a relatively short period (1 year), due to scarcity of sea temperature data at WAP environments, especially in coastal areas, it represents an important source of baseline information, that along with future high-resolution temporal observations on local temperature, will help us to better understand and forecast climate impacts on Antarctic marine organisms and ecosystems.

## ACKNOWLEDGEMENTS

The authors would like to thank F Beaujot, L Novoa, D Bravo, K Attard. L Rovelli, E DaForno, F Ressia and the INACH personnel at Yelcho Antarctic Research Station for help during field activities. We also thank Shane W. Geange for his assistance leading to an improved version of the manuscript. This paper contributes to the SCAR AnT-ERA programme.

### Funding

This research was funded by CONICYT/FONDECYT/INACH/INICIACION/#11150129. The funders had no role in study design, data collection and analysis, decision to publish, or preparation of the manuscript.

### Grant Disclosures

The following grant information was disclosed by the authors:
CONICYT/FONDECYT/INACH/INICIACION/: #11150129.

### Competing Interests

The authors declare there are no competing interests.

### Author Contributions

- César A. Cárdenas conceived and designed the experiments, performed the experiments, analyzed the data, contributed reagents/materials/analysis tools, wrote the paper, prepared figures and/or tables, reviewed drafts of the paper.
- Marcelo González-Aravena conceived and designed the experiments, analyzed the data, contributed reagents/materials/analysis tools, wrote the paper, prepared figures and/or tables, reviewed drafts of the paper.

- Pamela A. Santibañez analyzed the data, contributed reagents/materials/analysis tools, wrote the paper, prepared figures and/or tables, reviewed drafts of the paper.

### Field Study Permissions

The following information was supplied relating to field study approvals (i.e., approving body and any reference numbers):

The study was conducted under the permit 806/2015 granted by the Chilean Antarctic Institute (INACH).

### Data Availability

The raw and mean daily temperature values recorded at each site, and R code for the fitted generalized additive models and the method of finite differences for PY1, have been provided in the Supplemental Files.

### Supplemental Information

Supplemental information for this article can be found online at http://dx.doi.org/10.7717/peerj.4289#supplemental-information.

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
