# Peer review of "The importance of local settings: within-year variability in seawater temperature at South Bay, Western Antarctic Peninsula"

_PeerJ, doi:10.7717/peerj.4289_

## Round 0.1 · original submission · Major Revisions

I have now received three reviews of this manuscript. Although two of the reviewers recommended only "minor" revisions, their comments and suggestions are much more extensive than their recommendations would suggest. The third reviewer recommended rejection, yet that reviewer's comments echoed the comments of the other two reviewers.

Splitting the difference, therefore, I am returning the manuscript to you for major revisions. I expect that you will address all of the comments of the three reviewers, modifying the manuscript appropriately, and indicating where you disagree with the reviewers. I emphasize the following points:

1. One of the reasons to use a +3 degrees treatment is because that temperature reflects the endpoint of an expected trend in near-surface seawater temperatures. Although I agree with you that it is very important to account for natural variabilty in experimental design, and that inshore temperatures in places may already exceed +3 C at times, your dataset does not cover sufficient length to estimate even near-term (decadal) trends. The framing (introduction) of the ms. should distinguish between within-year and across-year variability, and the discussion should include comments on the contextual limitations of time-series data that cannot be used to estimate trend.

2. The most critical reviewer points out that there really is no hypothesis tested in the ms. Although in my opinion, the lack of a hypothesis is not grounds for rejection, you do need to be much clearer about the descriptive scope of the study and the limitations of the conclusions you can draw from a short-term, descriptive study for designing future experiments.

3. There are many experiments around the world that have worked with temperature variability and CTmax in designing experiments to examine responses of organisms to expected climate change. The diversity of taxa and sites used in these studies indicates the importance of the topic, and your review of the literature should include a representative range of them. In addition, there are a number of ongoing, long-term research programs in Antarctica, including on the WAP (e.g., the US Palmer LTER site, in addition to those referenced by the reviewers). You should look at their data and use it to better inform the context and interpretation of your findings.

4. Finally, the manuscript needs careful editing by a native speaker/writer of English. There are numerous grammatical errors (mis-matches of tense, case) and inappropriate word-usage throughout the ms.

Thank you again for submitting your manuscript to PeerJ. I look forward to reading your revision.

·

Basic reporting

The paper is well written, clear in the aims, results and in the take home messages. The use of English is correct and the language intelligible. The MS structure conforms to what is expected by PeerJ and the kind of research performed well fall within the Journal aims.

The most important ‘issue’ is that the results are based on local findings recorded in an area, i.e. the WAP, which is extensively studied by the scientific community in terms of climate change. Authors report for the first time (and as far as I know this is really the first time that such data are reported) that in shallow areas summer temperature increase already match levels that at the moment are only predicted to occur in the future. The article is logically and correctly based on this. However, I doubt that there are no other research programs in this area monitoring sweater temperature, at least in deeper waters (e.g. based on the use of oceanographic moorings, see for example here: http://www.soos.aq/activities/soos-at-sea/moorings) and these should be cited/commented.

I would therefore improve the Introduction by placing their study in a wider international and geographical context, and by better explaining and characterising the uniqueness of this dataset (i.e.: “this is the first of ever record of shallow water temperatures recorded in the WAP showing this this and this” or “despite there are several research stations recording seawater temperature in the shallow in this area (say where these are, if any) this is the unique yearlong record available showing this increase”…or “this is so far the unique available temperature record showing this increase”). These sentences will help the reader in understanding the scale of their finding and, in general, will underline since the introduction the importance of studies at the local scale (not only the big picture is important!) as well as the desperate need of more observation in the shallow which are urgently needed.
At the moment I consider the Introduction too brief (although correct) and I would suggest adding these details to complete the background information needed to properly introduce the topic.

The title, despite being clear in reporting the topic of the paper, does not contain any indication that the focus is the Antarctic and, since the fact that “treatments might be just controls” has been documented only here, it would be more correct to adding this information in the title as well in order to give a complete overview of the topic since the beginning.

Introduction, line 45. It is true that Antarctica has been considered one of the most stable environments on Earth but we are now well aware that major changes are occurring in some areas, such as the WAP, as well as that not all areas are warming up at the same speed. Some, such as the Ross Sea, are not warming up at all and show instead a reversed trend. Again, I would suggest completing this brief outline of the current Antarctic climatology taking into account (and referencing) this variability at the continental scale. This info will complete the findings reported in this paper and will underline even more the need to design site-specific thermal experiments in the future.

Introduction, lines 62-63. This sentence is not clear: what do Author mean with “to examine water temperature directly from observations”? Here in situ instrumental records are reported, which are (for sure) not commonly available for most inshore Antarctic environments. On the other hand, in situ measures of temperature are currently and constantly registered by a number of moorings deployed at sea all year-round in off-shore areas (see above link to SOOS). If their intention is to compare the accuracy and reliability of in situ measures vs indirect ones, e.g. those that can be obtained from satellites or remote sensing, this concept has to be better explained. Another point is the general lack of data in form of long-term series…and this is true both for shallow and deep environments. It is reported in Discussion (line 143) that long-term observational data sets in the WAP are limited to a few areas, but Authors do not mention which are these areas and if these data sets are available. It is not therefore possible to properly verify the uniqueness of their dataset.

The findings reported in this paper are about of two summers only and the real temperature trend (if any) is still unknown as well as the real interannual variability. This is implicit in the duration of their observation, but I think that other data about interannual variability for the WAP might be available. I would therefore invite Authors to comment about the possible causes of this variability and, again, if available, report about other projects from the same area.

All the literature cited is relevant.

Figure 1 is clear but I think it would be more useful if also bathymetric contours were included. I have no idea of the water circulation in the area but if any local pattern of is know (and relevant to explain any of the observed variability) This should be commented as well.

Figure 2 is not really necessary but, if considered relevant, add scale bar on Fig. 2B and depth information for both figures in the legend.

Figure 4 shows a series of ‘outliers’ for the station PY2 at -20 m (red dots) at the beginning of February. I have found no comments on this in the MS. Any hypothesis? It is possible that these data are due to instrumental failures or can be interpreted with a temporary introgression of colder waters? Does the general picture change if removed? Please comment on these records.

R scripts, raw data and input files have all been provided by the Authors and the scripts run. There is just a mismatch in the names of files compared to that given in the scripts (files now start with the “peerj-19157-“) but this is not an issue to run the scripts. However, in the final version of the paper this difference should be avoided. Figures do not appear to have been altered in any way.

Experimental design

The registered temperatures have been acquired to answer to a different research question (i.e. which are the responses of Antarctic sponge assemblages to climate change) from the one reported in the paper. Despite this, this is the first time to my knowledge that this topic is tackled in such a clear way and the finding that temperatures expected to occur in the next future are already met is of relevance to a variety of studies. For your info only: there is a specific SCAR group (ANTOS), which will provide protocols for monitoring nearshore environments and, hopefully, in the next future there will be a coordinated network of monitoring stations from which year-round temperature data could be obtained.

The analyses of temperature trends are all necessary, well done and effective in supporting the findings.

The experimental design is very simple and based on the use of HOBO data loggers at three different sites. Despite these loggers are of common use there may be differences in models and sensitivity. Please report manufacturer nominal data about sensitivity and specify the model used in order to allow comparison with other similar data in the future.

M&M line 77. Selected sites cannot be separated by 1.5 meters only. Please correct and report max and min distance between these sites. A scale on Fig. 1 would also help in figuring out the distance between these sites.

Validity of the findings

The data produced in this contribution are of relevance for the whole scientific community working on the thermal tolerance of Antarctic fauna as they indicate that the design of such experiments has to be carefully reconsidered and adapted to local conditions and temperature trend (of which a good knowledge is now required in advance!).
Since the WAP and the Ross Sea, for example, do show opposite temperature trends, with the latter not warming up at all, temperature experiments even on the same species would have to be modified and adapted in order to expose the specimens to temperature that are outside the range of those reached locally. In short, this means that temperature ranges cannot be ‘generalized’ in a ‘one size fits all fashion’ and that the same species might have populations, which have different thermal tolerances.
All these facts increase the complexity of thermal experiments, which will have, from now on, to take into account this ‘newly observed’ variability This introduces a higher level of complexity in studies dealing with thermal tolerance and will represent a challenge for researchers. Given this, I think that this paper is really important.

I would suggest to mention the paper by Ashton et al., 2017 (Current Biology 27, 2698–2705, http://dx.doi.org/10.1016/j.cub.2017.07.048) where ‘dramatic’ changes at the population and assemblage levels were found on settlement panels constantly heated just at 1 to 2 °C above the surrounding seawater temperature. In their experiment the increased temperature has been maintained for about one year, i.e. animals were exposed to higher temperature for a longer time. These two experiments, i.e. Ashton et al., 2017 and the data here presented, both cast new light on this topic and increase our understanding of the effect of climate changes and the need of carefully design thermal experiments.

Reviewer 2 ·

Basic reporting

The manuscript requires more editing. I started to make notes, but pretty quickly gave up as the amount of grammatical issues appeared too large to be handled in a review. Instead, I suggest the authors hire an editor or have a native speaker edited the manuscript.

The figures appear well developed, data are shared.

Article structure is ok, but the article does not test any specific hypotheses.

Experimental design

The study does not test any specific hypotheses. It simply summarizes data from three sites (different location and depth). The site selection does not have any apparent logic (or is related to a hypothesis) and locations are not sufficiently described, i.e., there is no reason to assume that site conditions are comparable, what specific inference scope the sites represent, or what factors are responsible for differences among sites.

Statistical tests indicate whether temperature is constant throughout various times of year, but the results of significant tests are not described in the text. Thus, the results only show that temperatures vary with seasons.

Validity of the findings

A basic assumption appears to be that quantifying the speed of temperature change within the year will provide information about the ecosystem responses to slower temperature changes due to climate change. The authors provide no convincing evidence that this is the case and it appears to me to be unlikely (note that I am not a marine biologist). Instead, it is more likely that organism have adapted to the annual temperature patterns and consequently critical temperature values will vary throughout the year, making the basic assumption invalid. Similarly, the data provide no information that changes in temperature (warming rates) are increasing, making a discussion about “heat shock” pure speculation (lines 161+). Why is the difference in warming rate before and after October relevant, is the difference even significant? In the same context, the data provide no information which site conditions are indicative of higher and lower environmental variability (lines 175+).

Additional comments

The general argument that experimental conditions should reflect natural trends makes sense, but is not new. The data at this stage are not sufficient to support the conclusion of the study. A better description of the sites and longer measurement period may be necessary. Alternatively, the data can provide a baseline for comparison with future measurements.

Reviewer 3 ·

Basic reporting

The authors clearly and professionally report their methods and findings in the present manuscript in which they report the temperature variation at three sites around Dourmer Island, Palmer Archipelago, WAP over the course of one year to better understand local variation and inform experimental set point in climate change experiments.

The authors present multiple figures describing the temperature dynamics, however I think that some of the figures depicting the raw data measurements might be better suited to be included as supplementary material. It seems like the same data are depicted in multiple graphs and it is unclear to me as written, what the added value is for including each of these figures. Perhaps more detailed description and interpretation would help put these different figures into context. Otherwise, maybe consider including some of these figures as supplementary material. The map (Figure 1) is very helpful in putting the present study site into geographical context.

The premise that temperature treatments should be put into a local context of exposure is extremely important. However, I think that the relatively narrow focused area observed somewhat geographically limit how applicable these results will be to other studies outside of this geographic region. Maybe some additional information on percent and duration of sea ice cover might help other researchers better gauge how applicable the results in the present study are their region of interest. They cite Schram et al. 2015, who appears to have data mined the Palmer LTER site for additional sea surface temperatures (range similar to that reported here). Perhaps the inclusion of this type of freely available data could add some weight to the data presented here?

Experimental design

The research presented is within the scope of the journal and their research question is well defined, relevant and meaningful – particularly in light that it appears that not all studies take local conditions into account when setting experimental levels for climate change experiments.

Validity of the findings

While I agree that it important know something about the local exposure of your experimental organisms, this information is not always available and therefore I would recommend caution when discussing the temperature levels used in previous experiments. I also believe that the authors could perhaps include some additional discussion on why studies that expose organisms to different temperatures are/are not still informative in the light that organisms may briefly experience those temperatures.

The authors point out that for their one year of data, organisms were exposed to “temperatures of more than 2.5C for more than 10 days within a year and over 2.0C for more than 25 days, temperatures that have been used experimentally to test responses to acute heat stress” (lines 124-126). I am not convinced that this is sufficient exposure to call these temperatures a control temperature. These elevated temperature days do not appear to have all occurred within the same week, therefore this may still be at the extreme temperature range of what organisms experience. Perhaps some additional justification is needed here? As it stands, I do not agree that the data presented for this one year study is sufficient to make the case that all Antarctic organisms experience these temperatures often enough to call this another control temperature.

Additional comments

When listing citations, if not an exhaustive list, perhaps precluding the list with"e.g."

---

## Round 0.2 · Minor Revisions

This is a much improved manuscript, and all of my comments and those of the reviewers have been satisfactorily addressed. I have only a few additional comments that I would like the authors to attend to in a minor revision, as detailed in the attachment.

---

## Round 0.3 · accepted · Accept

This is a timely paper. Having just returned from the West Antarctic Peninsula, I have an immediate appreciation for its value. Thank you for working through all the revisions.